SOFTWARE

# CellMet: Extracting 3D shape and topology metrics from confluent cells within tissues

**Sophie Theis**[ID][1]*, **Mario A. Mendieta-Serrano**[1], **Bernardo Chapa-y-Lazo**[1], **Juliet Chen**[ID][1,2], **Timothy E. Saunders**[ID][1]*

**1** Centre for Mechanochemical Cell Biology, Warwick Medical School, University of Warwick, Coventry, United Kingdom, **2** London Centre for Nanotechnology, Department of Cell and Developmental Biology, UCL, London, United Kingdom

\* sophie.theis@warwick.ac.uk (ST); timothy.saunders@warwick.ac.uk (TES)

**Data availability statement:** The source code is available at https://github.com/ TimSaundersLab/CellMet and at

## Abstract

During development and tissue repair, cells reshape and reconfigure to ensure organs take specific shapes. This process is inherently three-dimensional (3D). Yet, in part due to limitations in imaging and data analysis, cell shape analysis within tissues has largely been studied in two-dimensions (2D), e.g., the *Drosophila* wing disc. With recent advances in imaging and machine learning, there has been significant progress in our understanding of 3D cell and tissue shape *in vivo*. However, even after gaining 3D segmentation of cells, it remains challenging to extract cell shape metrics beyond volume and surface area for cells within densely packed tissues. To address the challenge of extracting 3D cell shape metrics from dense tissues, we have developed the Python package CellMet. This user-friendly tool enables extraction of quantitative shape information from 3D cell and tissue segmentations, including cell face properties, cell twist, and cell rearrangements in 3D. Our method will improve the analysis of 3D cell shape and the understanding of cell organisation within tissues. Our tool is open source, available at https://github.com/TimSaundersLab/CellMet.

## Author summary

The advent of machine-driven segmentation has made it possible - in a timely manner - to access complex 3D data quantitatively. Yet, the software tools to extract useful metrics to describe cell behaviour in dense tissues remain limited or not easily accessible. To meet the need for an accessible tool for extracting 3D cell metrics from cells within confluent tissues, we have developed CellMet. This tool analyses cell shape and organisation based on 3D segmentation. By extracting morphometric and organisational data in 3D, CellMet opens new avenues for better understanding cell shape within the tissue environment. This has a broad range of potential impacts, including in deciphering the underlying principles of morphogenesis. CellMet is free and aimed at being accessible to non-experts while also offering versatility to more advanced users.

https://pypi.org/project/CellMet/. Data acquired for this manuscript are available at https://zenodo.org/records/14177750.

**Funding:** ST, JC and TES are funded by UK Research and Innovation (EP/W023865/1). MAM-S and TES are funded by BBSRC Responsive Mode Grant (BB/W006944/1). BC-y-L and TES are funded by Warwick startup support. J.C. is supported by the MRC Doctoral Training Partnership in Interdisciplinary Biomedical Research at Warwick (MR/W007053/1) Kavli Institute for Theoretical Physics (KITP), University of California Santa Barbara, is funded by National Science Foundation (NSF PHY-1748958). This research was supported in part by grant NSF PHY-1748958 to the Kavli Institute for Theoretical Physics (KITP), where some of this work was initiated by ST and TES. The funders had no role in study design, data collection and analysis, decision to publish, or preparation of the manuscript.

**Competing interests:** The authors have declared that no competing interests exist.

## Introduction

Organs have specific shapes and sizes [1], with morphologies that ensure efficient function, such as the highly branched structure of the lungs enabling rapid gas exchange with the blood. Understanding how dense, three-dimensional (3D) organs form remains a major challenge in developmental biology [2–5]. This is in part due to the diversity of cellular processes that shape an organ, including division, migration, fusion, and extrusion [6–10]. Understanding cell shape dynamics is not restricted to development [11]. For example, injuries to epithelial tissues require wound repair to restore barrier function, whereby cells must reconfigure and populate the damage area [12,13] while undergoing long-ranged migration [14]. Finally, diseases such as cancer can induce shape changes in affected cells and tissues [15,16].

The physical shape, mechanics and functions of a cell are closely interrelated [17]. Further, the cell genetic programme is now known to respond to mechanosensitive inputs; morphogenesis and cell fate can influence each other during development and repair [18,19]. Such interactions can lead to self-organising pattern formation [20]. Cell morphological changes can be induced by local rearrangements involving acto-myosin modification [6,21]. By localising to specific surfaces, these processes can generate distinct cell shape changes [22,23]. However, effects of boundaries (e.g., the extracellular matrix (ECM)) and forces from neighbouring tissues can also impact cell and tissue shape [24–26]. Therefore, extracting quantitative information about cell and tissue morphology in 3D is an important challenge.

It has recently been shown that curvature can induce cell reshaping within tissues in 3D [27–29]. In epithelial monolayers, cells can adopt two characteristic shapes depending on the local curvature. In relatively flat tissues, cells form columnar (prism) structures, with the same cell neighbours on apical and basal surfaces. In highly curved environments (e.g., the anterior pole of the early *Drosophila* embryo), cells can undergo neighbour exchange between apical and basal surfaces [29], leading to a shape coined a *scutoid* [28]. Other processes, such as cell division [30–32] or orientated rearrangements [33] can also lead to significant variation in 3D cell shape. Recent theoretical work has highlighted the diverse array of possible cell shapes, even while maintaining confluency within tissues [34]. Quantifying these multi-scale processes is challenging, as both segmentation and quantitative analysis become increasingly difficult.

With the development of deep learning based pipelines for 3D cell segmentation, such as CellPose [35] or CartoCell [36], the capacity to analyse cell shape and how cells organise within a tissue in 3D has substantially increased. Although segmentation is becoming increasingly more accessible, the analysis of such segmentation still typically requires significant programming knowledge. Current tools to analyse 3D cell segmentation data have focused on describing cell morphology [37,38], employing methods such as spherical harmonics [39]. These approaches typically do not account for the cell neighbourhood or tissue environment, yet such information is critical in understanding 3D tissue morphogenesis. Current tools to analyse spatial rearrangements require scripting [40,41]. 3D cell shape analysis of plant systems (e.g., MorphographX [42]) is comparatively more advanced, in part due to the slower dynamics and high structural organisation of plant cells and tissue. Further, plant cell walls are typically more apparent, improving segmentation. In animal systems, cell shape is more malleable, facilitating frequent cell neighbour exchanges and changes in connectivity. Therefore, having a user-friendly tool that can analyse cell interfaces and neighbour exchange in 3D has broad potential application.

Here, we introduce CellMet to address these challenges of extracting quantitative cell shape metrics, which retain the structure of the cell, especially within densely packed tissues. CellMet enables quick and reliable extraction of relevant morphological measures from

3D cell segmentations, including cell morphometrics and face behaviour. We apply Cell-Met to a range of biological systems to demonstrate its versatility. Brief details of the specific experimental systems are provided in S1 File. 3D cell metric information can be powerful for addressing questions related to tissue material properties [43,44], inferring forces *in vivo* [45] and comparing with predictions from vertex models, for example. [46,47]. CellMet is written as a free Python package available at https://github.com/TimSaundersLab/CellMet and through Pypi. The structure is such that the tool is readily usable by non-experts but freedom is available for more advanced users to optimise to their specific needs.

## Design and implementation

### Ethics statement

All work on Zebrafish was approved by the University of Warwick animal welfare and ethical review board (AWERB, code 77 20-21) and adhered to the Animals (Scientific Procedures) Act 1986, and Home Office ASPeL regulations for animal work.

### CellMet description

CellMet is designed for the analysis of 3D cell segmentations. It is not aimed at analysing 2D segmentations. When considering how cells pack within dense tissues it is necessary to extract broader information, such as how edges and faces vary in morphology, and the number of cell neighbours (connectivity) at different locations within the tissue. CellMet significantly simplifies extraction of this information from 3D segmentations.

Several output files are generated by CellMet at different steps. CellMet works by decomposing the labelled image into single cell images for faster processing. To optimise digital storage, these cell images are stored as NPZ Numpy's compressed array format. Alternatively, each cell object can be exported as .obj or .ply files, which is compatible with other 3D rendering software such as Blender or Paraview (Fig 1).

CellMet is written in Python and has been developed on Gnu/Linux. It has been tested on Windows and Mac operating systems. Code and installation can be found here: https://github.com/TimSaundersLab/CellMet. We provide example Jupyter notebooks to explain CellMet use.

### Pre-process and prerequisites

CellMet requires 3D segmented images to perform the analysis. Here, we refer to a segmented image as a labelled image where each pixel value corresponds to a specific identified cell (ID). As CellMet does not correct for mis-segmentation (missing segmented cells or over-segmentation, i.e. two IDs for one cell), this needs to be done prior to analysis. There already exist tools for such analysis, and are included in packages such as Napari or FIJI. However, within CellMet it is possible to filter the extracted metrics to remove outlier cells; e.g., removing cells with very large volume, which is likely two cells mistakenly combined.

### Decomposing cell, edge and face segmentation to a vertex - like structure

To decompose cells into faces and edges, we need to determine the local neighbourhood of each cell, *i.e.,* identifying cells in direct contact with the chosen cell. This then allows us to measure more detailed 3D morphological metrics, such as area of contact between two cells or the tortuosity of the contact between three cells (Fig 2). With this framework, we can quickly extract information about cell edges and faces, which can be utilised to quantitatively assay 3D changes in cell shape and cell neighbourhood.

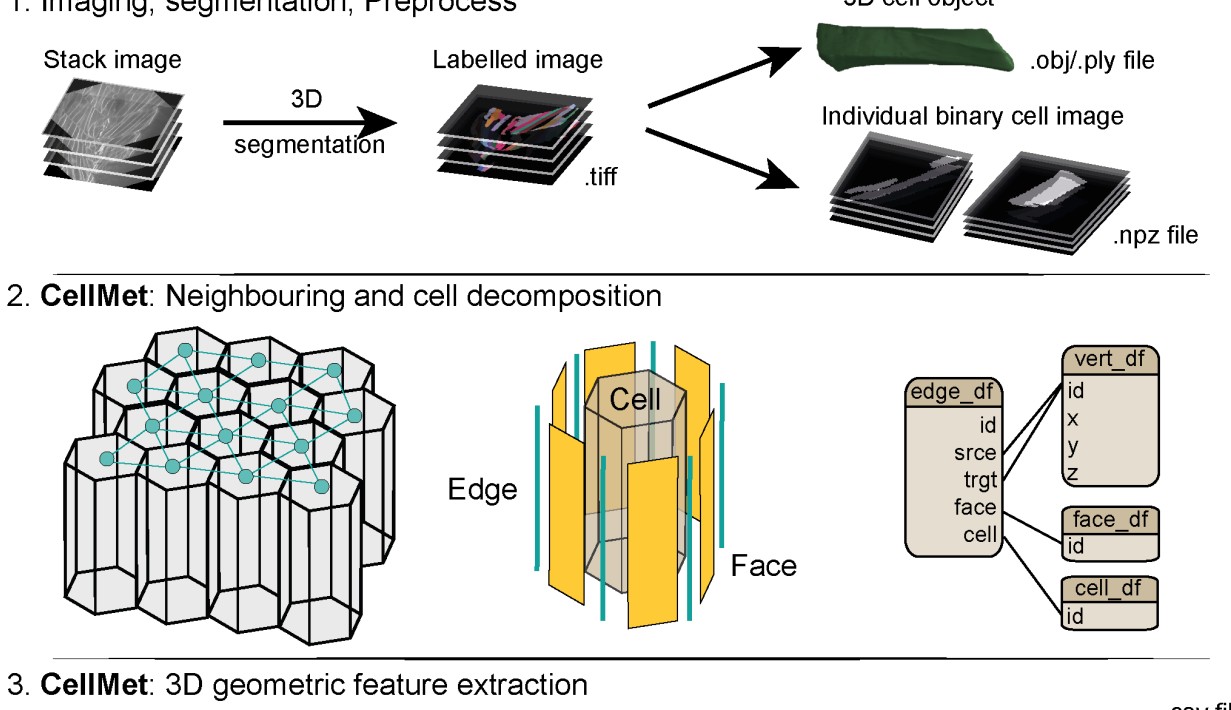

**Fig 1. CellMet pipeline. Preprocess**: Acquisition of a Z-stack. 3D cell segmentation is performed to obtain a 3D labelled image. **Prerequisite**: Create single cell files for faster analysis. Possibility to create 3D object file for 3D rendering. **CellMet analysis**: Analysis of the shape of cells and neighbour relationship. **Export**: Results are stored in .csv files.

Our approach decomposes the cell into smaller, more manageable structures. We organise this information using a vertex-based representation of morphology, a data structure that models cells, their faces and edges as interconnected elements. This approach is analogous to building a polygon mesh representation of the cell, where our data structure enables us to identify each polygon (face) with each cell. The edge table is used to link informations between cell and face tables. This approach enables us to store cell shape information as a collection of straight line polygons. To preserve finer geometric details such as shape and straightness, additional csv files are saved which contain pixel-level information for each face, edge and cell (Fig 1 and S1 FigA). To efficiently manage the relationships between the geometric elements, we internally use a half-edge vertex model structure [48].

To determine cell neighbours, we assume that cells are in close contact with each other. We use the one-cell binary image and dilate by 1 pixel. We then multiply this image by the labelled image. The remaining pixels then belong to the neighbouring cells and their values correspond to the cell ID. Our approach requires cells to be tightly packed; CellMet is not currently easily applicable to tissues with large inter-cellular spaces. We use these informations

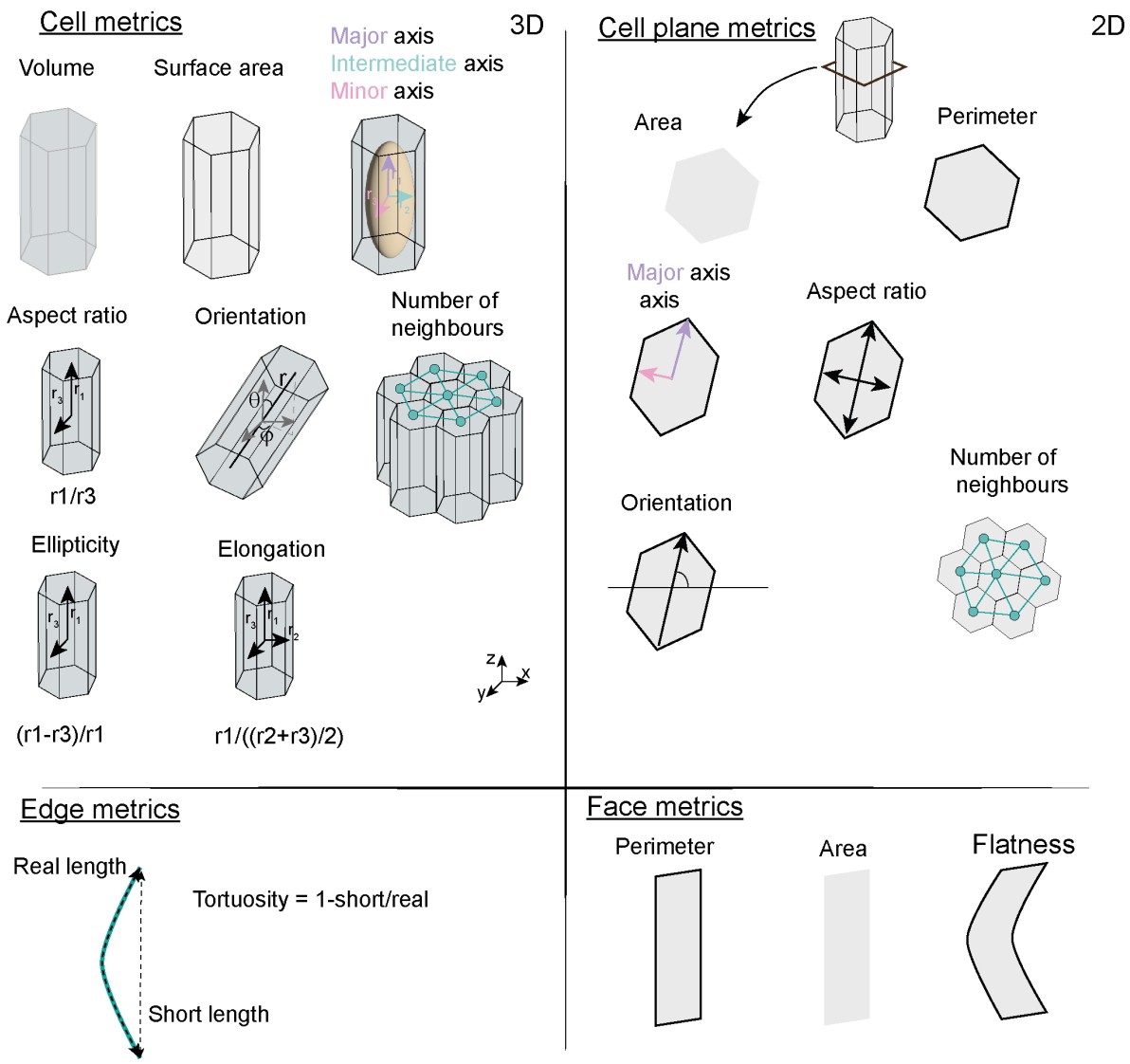

**Fig 2. CellMet metrics examples.** 3D and 2D cell visualisation of metrics measured in CellMet for cell (top left), cell plane (top right), edge (bottom left) and face (bottom right).

to construct a Region Adjacency graph (RAG) or connectivity graph: $G(V,E)$, where $v_i \in V$ represents the $i$–th cell and $e_{ij} \in E$ the link between two cells $(v_i, v_j)$.

The second step consists of determining cell edges. At vertices, these edges are shared between three cells, whether for prism or scutoid shapes. For scutoid-type cells, edges are still defined by three cells; however, edges do not necessarily follow the entire length of the cell from apical to basal surfaces. For each cell, all possible combinations of three cells are determined. These three-cell binary images are then multiplied together; if they are not neighbours then the output image is blank. Otherwise, the identified non-zero pixels correspond to the junction between those specific cells (S1 FigB).

Finally, we determine the surface contact between each pair of neighbours. For each pair, we multiply the binary images together. The non-zero pixels correspond to the interface surface between the cells. Though this approach is simple, it can only determine cell faces and edges if the neighbours are in closed contact. CellMet partially extracts faces and edges for cells at the edge of the segmentation (S2 FigB).

## Cell analysis

Utilising our data tables, we can extract the morphological metrics for a specific cell. For ease of measurement and to ignore differences in pixel size (especially in z-axis vs x/y-axis), images are reshaped into an isotropic pixel size. Volume and surface area are measured by multiplying the number of pixels on the inside or on the outer surface of the cell by the voxel or pixel size respectively. We note that this approach of area measurement tends to overestimate the surface area [49,50]. To find the centreline (or Skeletal Line), we use the average position of each plane from top to bottom. To take account of the tilt of the cell, which can lead to an error in calculating the centreline, we reorientate the cell so that its major axis align along the z-axis before finding the Centerline. We then rotate the Centerline back into the original space. To avoid unnecessary rotation, only cells that are tilted more than 30° to the z-plane are reoriented (S3 Fig). Orientation of the cell is measured with respect to the three different planes: xy, xz and zy.

**Cell straightness and sphericity.** From this data, we can extract two important metrics: (i) the shortest straightline distance from one end of the cell to the other, $l_{short}$; (ii) the actual distance along the line, which includes contours $l_{real}$. Cell straightness is calculated from those lengths as: $t_s = 1 - \frac{l_{short}}{l_{real}}$. So, within this definition $t_s = 0$ corresponds to a perfectly straight cell edge.

Cell sphericity is obtain following the definition by Wadell [51]:

$$\Psi = \frac{\pi^{1/3}(6V)^{2/3}}{A} \qquad (1)$$

where A and V are the surface area and the volume of the cell respectively. We note that there are other definitions of sphericitiy, and CellMet can be easily edited to output the appropriate measure.

In addition, CellMet can extract basic 2D shape metrics at each z-plane to supplement the 3D information. For each plane CellMet can measure the area, perimeter, orientation, anisotropy, and major axis (Fig 2). We emphasise that CellMet is focused on 3D analysis, but providing 2D metrics can be useful for comparisons.

**Face analysis.** Cell edge and face orientation is important in a range of biological processes [52]. CellMet can extract the length and the angle of cell faces across each z-plane (Fig 2). To facilitate the analysis of face shape, we use singular value decomposition (SVD) to project the face in a 3D space onto the best 2D fitting plane [53]. This enabled us to quickly extract information about the face geometry. We note that this method can be imprecise if the face is highly deformed or bent; typically, this would lead to area and perimeter being underestimated. Flatness of the face is measured as the variance of the third principal component of a PCA (if $f = 0$ => face is flat; if $f > 0$ => face not flat) [54,55].

### Hardware

For image processing, data analysis and visualisations we used a workstation running Ubuntu 20.04, with 64GB RAM, Nvidia A6000 GPU 48GB and processor Intel Xeon w2235 CPU 3.80GHzx12core.

### Generating artificial training images

To generate an artificial segmented tissue we used the Voronoi diagram from the Scipy Python package to obtain a 2D cell pavement. This pavement was duplicated on several slices to create a 3D labelled image. Seeding point density was varied to obtain more or less cells as required.

For image size comparison, we generated an image with $500 \times 500 \times 150$ pixels and with $\sim 70$ cells. Using the scale function in ImageJ, we changed the image size to $250 \times 250 \times 150$ and $1000 \times 1000 \times 150$ pixels. This enabled us to compare the same cell topology with different image size.

For cell number comparison, we generated an image with $500 \times 500 \times 150$ pixels with $\sim 150$ cells. We then removed border cells to reach two new datasets with $\sim 70$ and $\sim 30$ cells, keeping the same average cell size.

## Results

### Applications

CellMet quickly characterises cell shape in 3D from individual cells that are tightly packed within a tissue. Here, we discuss its applications to a number of example systems and highlight strengths and weaknesses of our approach.

The shape analysis of 3D cells can be divided into two categories: first, data obtained from each cell individually; second, information about neighbours (Fig 2). All outputs from CellMet are stored in .csv files (Fig 1).

***Drosophila* gastrulation.**   Building mechanistic understanding of tissue morphogenesis - for example, during development or repair - requires knowledge of cell shape within the tissue context. Here, we demonstrate how CellMet can extract general cell properties as well as edge positions and surface contacts with neighbouring cells during *Drosophila* gastrulation (Fig 3). Focusing on the number of neighbours, we observe a significant increment in 3D connectivity during mesoderm invagination (Fig 3). The sharp increase is not observable by considering the apical surface alone, even though cells undergo a substantial change in size [56]. As we do not consider boundaries, the neighbour number for cells at boundaries is reduced (Fig 3D).

We can also focus on individual cell shapes. As gastrulation occurs, cells need to rearrange in 3D to stay tightly connected while adjusting for the spatial changes. As a consequence, complex 3D cell shapes, such as scutoids can emerge [27,28]. Using our single cell analysis, it is straightforward to identify cells that undergo neighbour rearrangements along their apical-basal axis (Fig 4A). We can then extract quantitative information about how cell morphology can alter along its apical-basal axis (Fig 4B).

***Drosophila* heart.**   We have extended our analysis to different tissues and model organisms with various organisations. While scutoid geometries have received significant attention, there are other means of 3D cell packing. An example is during *Drosophila* heart closure, where neighbouring cardioblasts must precisely align [58,59]. We applied CellMet to a stage 16 *Drosophila* embryo heart (Fig 5A). Using our surface deconstruction, we can easily calculate the surface overlap between neighbouring cells. Further, we calculated the connectivity graph (Fig 5A), which reveals information about the precision of cardioblast alignment,

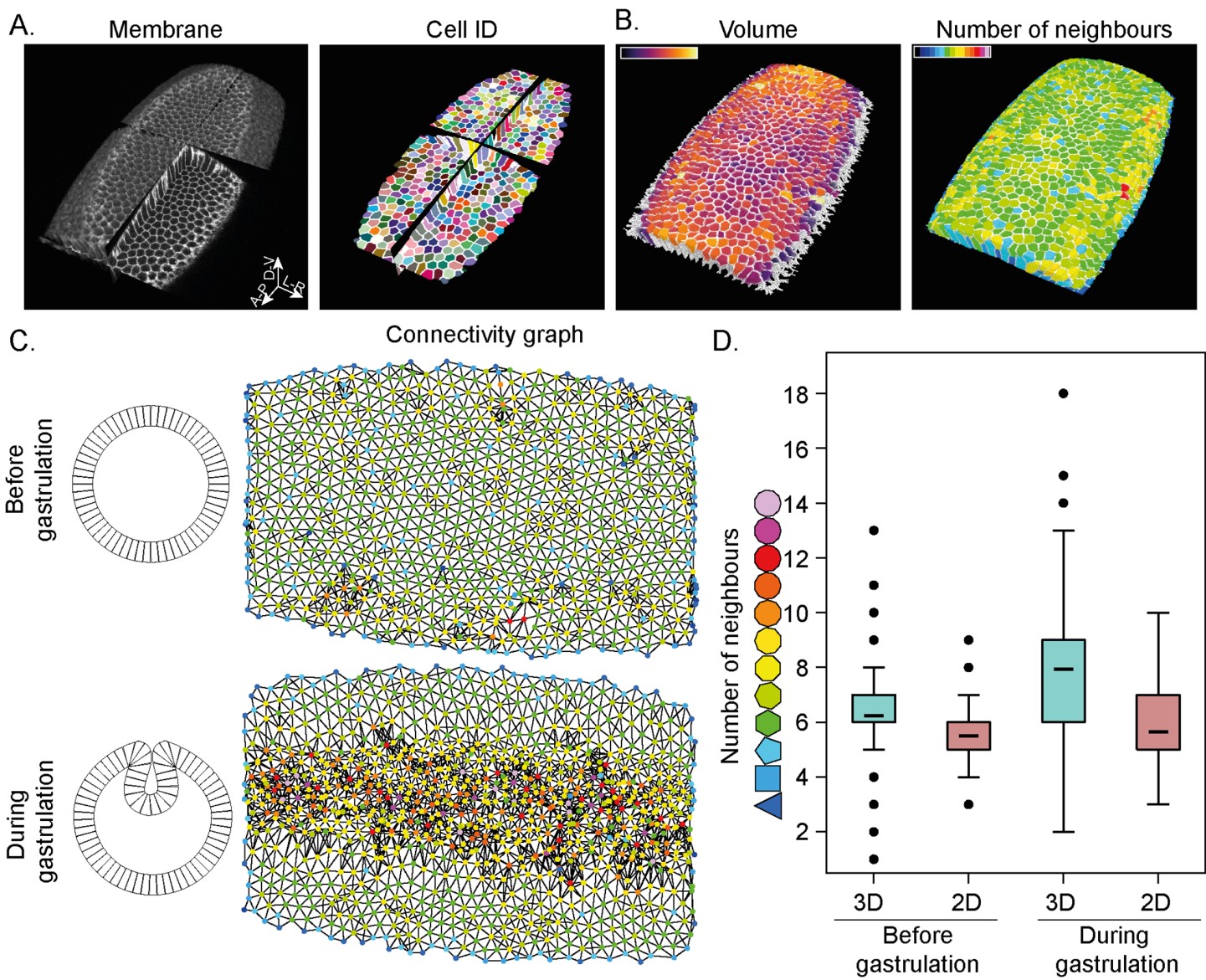

**Fig 3. Application to *Drosophila* gastrulation.** A. Multi orthoview of PH-mCherry embryo during mesoderm invagination from Gracia [57] (left) and cell identity (right). B. Colour coded cells show volume (left) and neighbour number (right) in the embryo. C. Scheme showing tissue invagination (left) and the corresponding connectivity graph from A (right). The top row is before gastrulation and bottom row is during gastrulation. Node colour represents the number of neighbours and edge colour represents the link between two cells. D. Violin plot of the number of neighbours from C. Measures made in 2D at the apical cell surface (brown) and in 3D (cyan). $n \simeq 900$ cells from one embryo.

especially in the absence of specific cell fate markers. This provides a much quicker method to assess cell alignment, compared to human-annotated approaches [60]. Such data is important in quantitatively testing models of cell alignment [61].

**Zebrafish myogenesis.** During morphogenesis, zebrafish form muscle segments (myotomes) along their anterior-posterior axis (Fig 5B left). These develop into a characteristic chevron morphology, where muscle cells have to fit closely together [24] in order to synchronously contract to make the fish move.

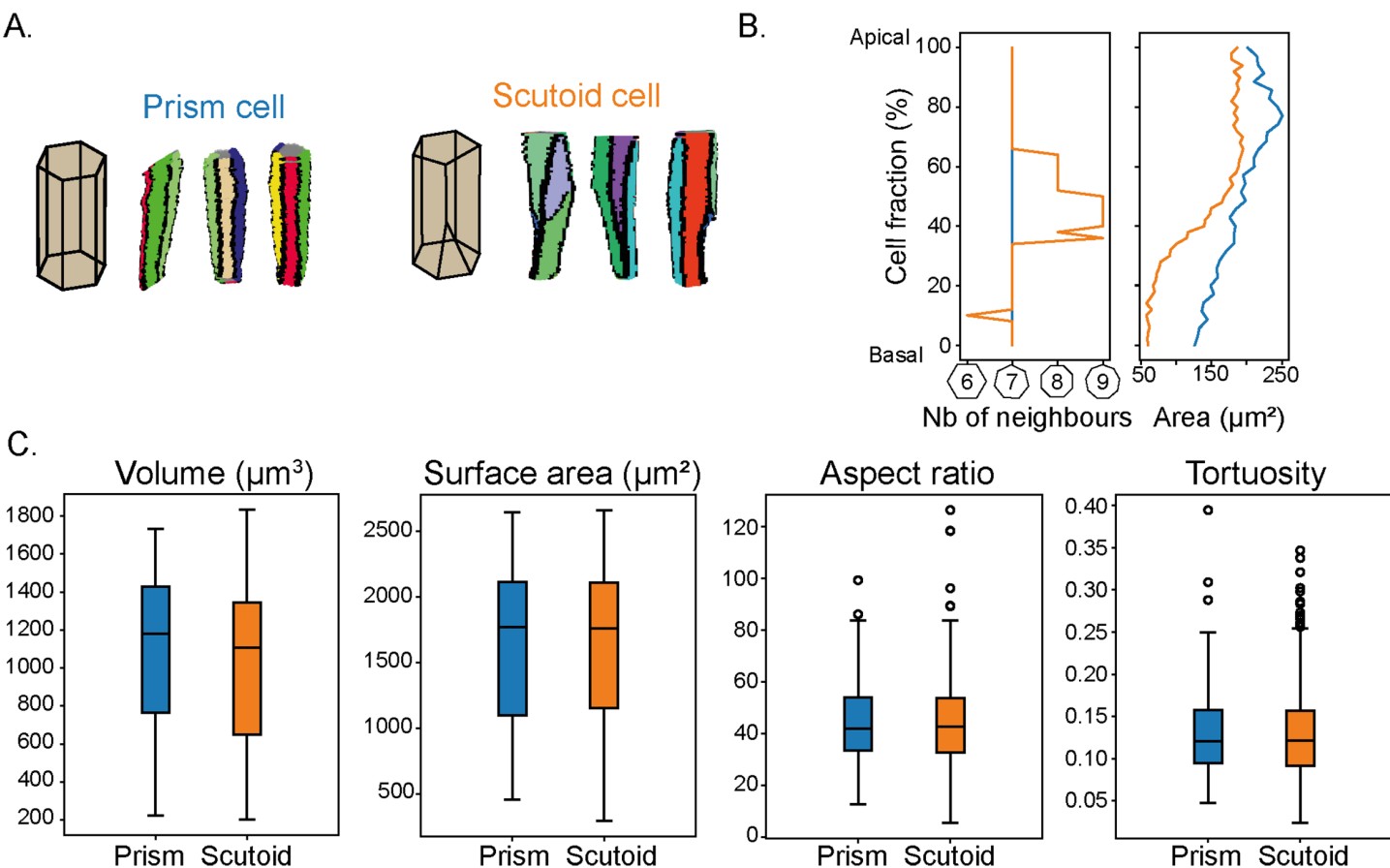

**Fig 4. Prism and scutoid cell shape.** A. Colour coded face in prism (left) and scutoid (right) cells. Edges are in black. B. Change in number of neighbours and area along cells. Blue and orange represent a prism and scutoid cell respectively. C. 3D morphology measures for different cell shapes extracted from CellMet.

Using CellMet, we can quickly and precisely highlight contact edges and surfaces along the myotome (Fig 5B right). By analysing cell shape in 3D, we were able to identify a broad range of cell shapes. We observed both straight and twisted cells. In the latter case, cells appear to rotate relative to their neighbours. By tracing the edges of such cells (colour coded dots in (Fig 5B right)), we can quantify their rotation around the cell centre. Thus, CellMet enables analysis of complex cell morphologies that are distinct from 2D behaviours.

**Organoid system.**   Organoids have become powerful tools for understanding organ development, including in humans [62]. Yet, detailed analysis of cellular morphodynamics in 3D within organoids remains very limited [63]. Here, we demonstrate that CellMet is a powerful tool for extracting relevant morphological information from organoids.

We use an organoid culture, termed neuruloid, which mimics dynamics of stem and progenitor cell populations in the posterior embryo [64]. After 48 hours, the cells become organised in a multilayered epithelium, making it difficult to quantify cell shape (Fig 5C left). Utilising a trained model in CellPose, we were able to segment the cells in 3D within the multilayered tissue (Fig 5C centre). With CellMet, we could then identify the cell interfaces. From this, we built the cell connectivity graph to understand how cells are organised within the tissue. In tandem, we extracted cell shape properties, such as cell sphericity. We see that there is correlation between cell neighbour number and cell shape (Fig 5C right).

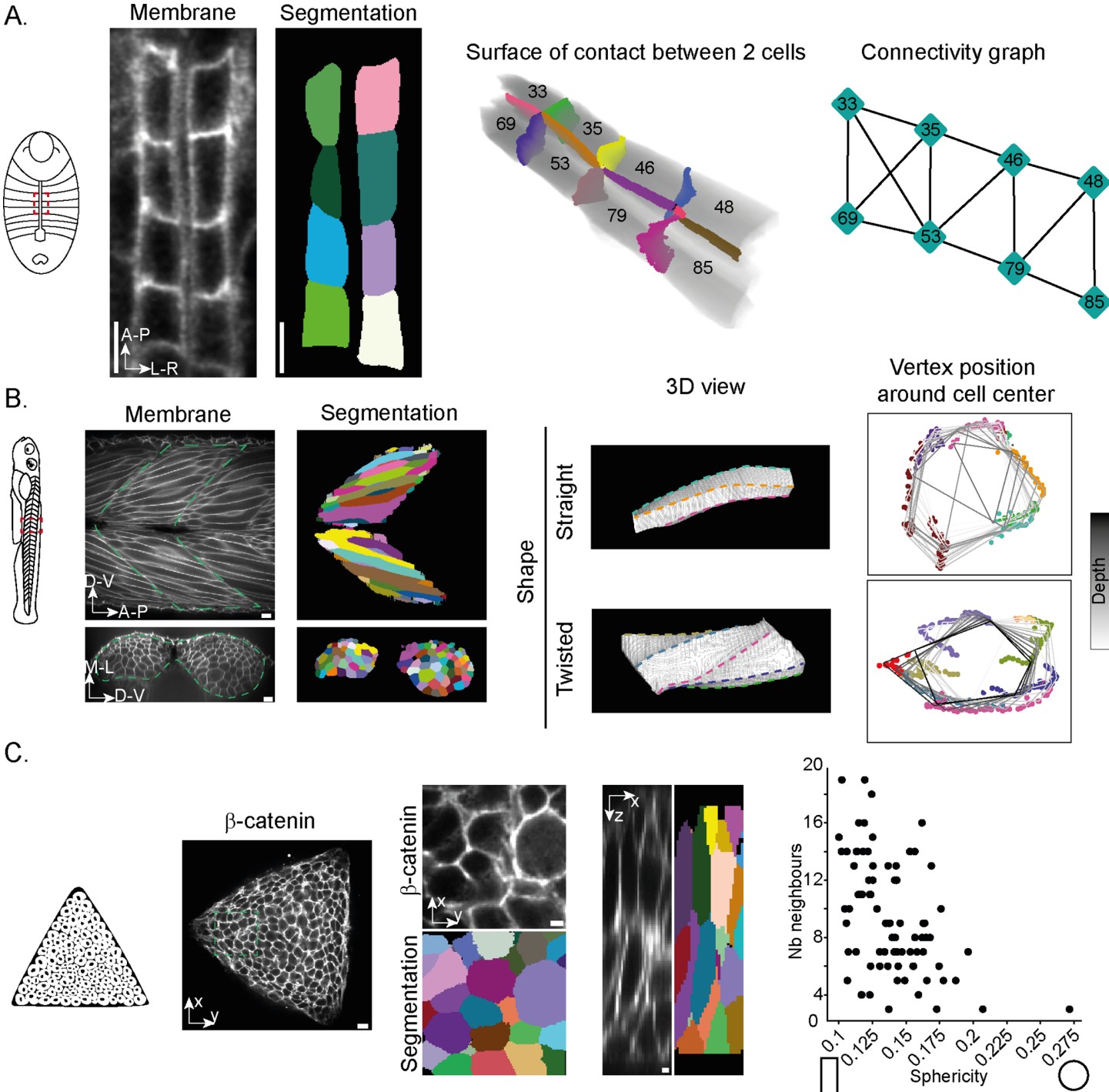

**Fig 5. Application of CellMet to different tissues.** A. (Left) Schematic of the *Drosophila* embryo from a dorsal view, with anterior up. The red dashed region corresponds to the images to the right. (Centre) Heart cells in a stage 16 *Drosophila* embryo, just as the heart lumen forms with segmentation from CellPose. (Right) Connectivity of the heart cells, demonstrating the cell alignment. Numbers correspond to cell labels ID. B. (Left) schematic of the zebrafish embryo, highlighting the myotome region with red box. (Centre) Myotome in 48hpf zebrafish embryo, with segmentation from CellPose. Green dashed line shows the chevron form of the myotome. Bottom shows cross-sectional view. (Right) 3D visualisation of muscle fibres showing the different cell morphologies. For the twisted cell, we can trace the rotation along the cell long axis. Vertex colours are the same as edges showed in 3D. C. (Left) Schematic of organoid culture constrained to triangular domain. (Centre) Organoid culture at 48 hours, with CellPose segmentation. Views shown in x-y and x-z axes. (Right) Comparison of cell sphericity with cell connectivity. Scale bar: A: 5$\mu m$; B and C: 10$\mu m$.

Overall, Fig 5 shows that CellMet enables comparison of different cell metrics in 3D, which provides important information for understanding how densely packed tissues are organised.

## Performance testing

To verify the efficiency of CellMet, we generated a synthetic epithelium with a known number of cells (Design and Implementation). We created variants of our synthetic data with different sizes of the same images. Unsurprisingly, larger images increased the time required for analysis. Doubling the image size increased the execution time by a factor of six (Fig 6A). Further, larger images led to memory issues, especially when using parallel execution to quicken the analysis. The execution time also increased with cell number (Fig 6B). The main cause for this slow-down was due to the face-analysis step, as more cells had a large number of neighbours.

## Comparison with other software

There have been a range of powerful tools developed to analyse cell shape. These have different strengths and weaknesses; comparing such software methods is not straightforward as most have been built to address particular problems.

Here, we compare a range of packages used commonly to analyse 2D and 3D segmented data sets; Table 1.

CellMet includes a broad range of metrics, that cover both the cell morphology and tissue connectivity. This contrasts with most existing tools that have more specialist functionality,

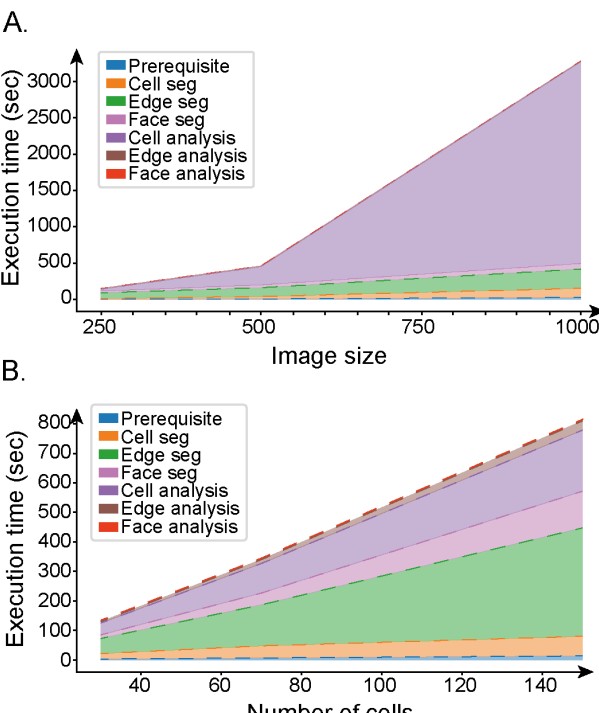

**Fig 6. Performance.** Execution time on synthetic data using a computer with $n_{core}$ = 10, and RAM = 64 GB. A. Relationship between the size of the image (*i.e.* number of pixels) and the duration of execution. B. Cumulative execution time according to the number of cells in the image. Each subprocess is colour coded (legend), to show the specific time demands.

**Table 1. Comparison of software packages for analysing segmented data.**

|  | CellMet | CellProfiler [65] | CartoCell [36] | Griottes [66] | iPS cells [37] |
|---|---|---|---|---|---|
| 2D |  | ✓ |  | ✓ |  |
| 3D | ✓ |  | ✓ | ✓ | ✓ |
| Single cell |  | ✓ |  |  | ✓ |
| Cell in epithelium | ✓ | ✓ | ✓ | ✓ | ✓ |

e.g., Griottes [66] focuses on network connectivity. See Table 2 for a brief overview of different software capabilities. We provide simple notebooks, which ensure CellMet is readily available to non-experts. Importantly however, the code is easy to edit, allowing users to customise to their particular needs.

## Availability and future directions

CellMet is a free Python package for extracting quantitative shape information about cells within dense tissues. It is available through PyPi and at https://github.com/TimSaundersLab/CellMet. CellMet is published under an open-source (GPLv3) license to encourage the sharing of the resources. We provide notebooks with examples showing its application, with example datasets [67]. CellMet will work on any segmented data with clear cell identification and sufficient packing.

## Limitations and constraints

CellMet is not suitable for analysing single cells, e.g., migrating cells with large protrusions [68]). There are more specialist tools available, e.g. [11] for such analysis. CellMet is relatively fast, with most analyses completed in under 10 minutes. However, it is not highly optimised,

**Table 2. Comparison of software packages aimed at extracting metrics from segmented data.**

|  | CellMet | CellProfiler [65] | CartoCell [36] | Griottes [66] | iPS cells [37] |
|---|---|---|---|---|---|
| Volume | ✓ | ✓ | ✓ | ✓ | ✓ |
| SurfaceArea | ✓ | ✓ | ✓ | ✓ |  |
| Aspect ratio | ✓ | 2D only | ✓ |  | ✓ |
| Cell solidity | ✓ | 2D only | ✓ |  |  |
| Cell orientation | ✓ | 2D only | ✓ | ✓ | ✓ |
| Cell eccentricity | ✓ | 2D only |  | ✓ |  |
| Fluorescence |  |  |  | ✓ |  |
| Number of neighbours | ✓ | ✓ | ✓ |  |  |
| Connectivity graph | ✓ |  |  | ✓ |  |
| Edges (decomposition) | ✓ |  |  | ✓(length) |  |
| Faces (decomposition) | ✓ |  |  |  |  |

and so slows down with very large images. In this case, data will need to be partitioned to make it more manageable.

We have design CellMet to analyse cells in a tightly packed tissue in 3D, either monolayer or multilayer, but there are a few limitations. The method used to determine faces and edges brings two major limitations. First, it is not suitable to study finer structures such as filopodia or other highly protrusive shapes that are only a few pixels across. Second, CellMet is not currently easily applicable to tissues with large inter-cellular spaces, as these regions impact the method to identify faces. Similarly, CellMet does not analyse cell faces at the outer boundaries. At present, the top and bottom of cells within a monolayer are not considered as faces themselves. Although CellMet provides 2D measurements, there are more powerful options available for such analysis (Tables 1 and 2).

The accuracy of the analysis is dependant on the quality of the segmentation. CellMet does not correct for mis-segmentation; this needs to be done prior to analysis. Data can be filtered based on given parameters to remove aberrant cells.

## Future directions

In principle, the output from CellMet can be combined with other forms of biological image data. For example, given the edge and face details, signal fluorescence and localisation changes over time can be quantified across specific spatial domains. This would be relevant for exploring how polarity factors alter their localisation during tissue morphogenesis [69], how morphogens form concentration gradients within dense tissues [70,71], and how the mechanosensitive protein Piezo1 changes its location and activity levels during organogenesis [72].

As this tool was developed to answer our needs, useful metrics may be missing. Our tool is designed to evolve to meet everyone's needs. Extensions can be implemented via GitHub, providing quick and easy access to new code. To extend the accessibility to CellMet, we plan to develop a Napari plugin [73].

The output of CellMet utilises vertex-like approaches to tabulate the results. This means the data structure easily permits a link between experimental observations and mechanical models of tissue morphogenesis [74]. This information will be important for testing models of how, for example, mechanical stress influences organ shape [75,76].

The addition of boundary layers will be an important next step. This will enable information from boundary constraints to be more rigorously integrated with cell shape changes. Further, in principle we can include extracellular spaces within the framework (essentially by treating such spaces as specially marked "cells"). The presence of such spaces within even densely packed tissues can play an important role in tissue morphogenesis [43].

In summary, we provide a computational framework for extending our understanding of tissue morphogenesis into three-dimensions. CellMet is aimed at general users; it does not require extensive coding skills.

## Supporting information

**S1 File. Experimental methods.**
(PDF)

**S1 Fig. Nomenclature for 2D and 3D cell decomposition.** A. Cell decomposition into faces, edges and vertices in 2D (top) and 3D (bottom). B. Minimal example of how edge is determined in 2D using CellMet.
(TIF)

**S2 Fig. Cell, Edge and Vertex detection.** A. Cell detection is based on the segmentation. Cells dilated by one pixel are generated to facilitate face and edge detection. B. Faces are found by multiplying the labelled image by one dilated cell. Non-zero pixel values correspond to the neighbouring cell ID. Cells at the edges will not have all their faces detected (bottom). C. From the list of neighbouring cells, we find all combinations of 3 cells and multiply these 3 dilated cells. Pixels remain when they form a junction, otherwise they are removed to indicate empty space.
(TIF)

**S3 Fig. Calculation of the Cell Centreline.** A tilted cell can lead to miscalculation of the cell centreline (left). If the major axis orientation is >30° to the z-axis, then CellMet reorients the cell (middle). The cell centreline is calculated and then put back to the original orientation (right).
(TIF)

## Acknowledgments

We thank James Briscoe, Guillaume Charras, Sarah Goodband and Tiago Rito with support in generating organoids and comments on the work. We thank members of the Saunders lab for useful input in the design of CellMet. We thank the Suzanne lab for sharing their data on *Drosophila* embryo gastrulation.

## Author contributions

**Conceptualization:** Sophie Theis, Timothy E. Saunders.

**Data curation:** Sophie Theis, Mario A. Mendieta-Serrano, Bernardo Chapa-y-Lazo, Juliet Chen.

**Formal analysis:** Sophie Theis, Mario A. Mendieta-Serrano.

**Funding acquisition:** Timothy E. Saunders.

**Investigation:** Sophie Theis.

**Methodology:** Sophie Theis.

**Project administration:** Sophie Theis, Timothy E. Saunders.

**Resources:** Sophie Theis, Mario A. Mendieta-Serrano, Bernardo Chapa-y-Lazo, Juliet Chen.

**Software:** Sophie Theis.

**Validation:** Sophie Theis.

**Visualization:** Sophie Theis.

**Writing – original draft:** Sophie Theis, Timothy E. Saunders.

**Writing – review & editing:** Sophie Theis, Mario A. Mendieta-Serrano, Bernardo Chapa-y-Lazo, Juliet Chen, Timothy E. Saunders.

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
