## [Decision Letter · Decision Letter 0]

31 Jan 2025

PCOMPBIOL-D-24-01768

CellMet: Extracting 3D shape metrics from cells and tissues

PLOS Computational Biology

Dear Dr. Theis,

Thank you for submitting your manuscript to PLOS Computational Biology. After careful consideration, we feel that it has merit but does not fully meet PLOS Computational Biology's publication criteria as it currently stands. Therefore, we invite you to submit a revised version of the manuscript that addresses the points raised during the review process.

Please submit your revised manuscript within 60 days Apr 02 2025 11:59PM. If you will need more time than this to complete your revisions, please reply to this message or contact the journal office at ploscompbiol@plos.org. Please include the following items when submitting your revised manuscript:

We look forward to receiving your revised manuscript.

Kind regards,

Virginie Uhlmann

Academic Editor

PLOS Computational Biology

Marc Birtwistle

Section Editor

PLOS Computational Biology

**Journal Requirements:**

3) Your manuscript is missing the following sections: Design and Implementation, Results, and Availability and Future Directions. Please ensure that your article adheres to the standard Software article layout and order of Abstract, Introduction, Design and Implementation, Results, and Availability and Future Directions. For details on what each section should contain, see our Software article guidelines:

https://journals.plos.org/ploscompbiol/s/submission-guidelines#loc-software-submissions

5) We notice that your supplementary Figures are included in the manuscript file. Please remove them and upload them with the file type 'Supporting Information'. Please ensure that each Supporting Information file has a legend listed in the manuscript after the references list.

6) Please ensure that the funders and grant numbers match between the Financial Disclosure field and the Funding Information tab in your submission form. Note that the funders must be provided in the same order in both places as well.

**Reviewers' comments:**

Reviewer's Responses to Questions

**Comments to the Authors:**

Reviewer #1: # CellMet: Extracting 3D shape metrics from cells and tissues

This paper presents a python software library for analyzing cells in 3D. I think the paper needs

major revisions for publication.

## Major general problems.

1st. The structure itself does not fascilitate understanding what the package is doing.

2nd. It is not clear to me what measurements are being made and what approximations they use.

3rd. The end of the paper says that the software is for "general users" which is too vague.

### Improving the structure of the paper

The authors reference a decent number of publications that use similar measurements and analysis. I know that these publications have made

their software available too. Something that could help would be a table with types of measurements used for analyzing epithelia vs software packages.

Say which features each software package supports or doesn't support.

I think this would greatly improve the readability of the paper and show what is novel about it. What about cell profiler or

CartoCell? They reference Scutoids, what about the analysis performed in that paper.

### Clear description of what measurements are, how they're calculated and what approximations are being made.

I did not understand how does a general blob of pixels get transformed into a table of faces, edges and vertexes representation.

Could this demonstrated with some very simple geometries? Eg. A sphere? Or if this is not applicable for a sphere, the authors should be clear about what it is applicable.

The description for generating surfaces seems rely exclusively on the overlap of 1px dialated cell-cell overlap. I could see this causing some problems with finer features.

The determination of edges looks at 3 clusters of cells. It seems like this isn't sufficient for 3D environments, for example scutoid cells.

"Our approach requires cells to be tightly packed;" Are there other requirements? There also is a statement about protrusions or processes. Do you have a specific metric that would let somebody eliminate this algorithm, or possibly refine their segmentations to be more suitable?

In figure 3, the connectivity has " The sharp increase is not observable by considering the apical surface" Maybe this could be illustrated with a cell that has neighbors that are not connected in an apical only analysis.

"Further, we calculated the connectivity graph (Fig 5A)" this is a good candidate to compare to other software packages that can create a connectivity graph. Does it exist? The only comparison is "This provides a much quicker method to assess cell alignment, compared to human-annotated approaches"

If this is going to be used as a general software package, it should explore the limitations in a more systematic and quantitative manner.

### Who are general users?

By addressing the first part, and the second part, this part should come out naturally.

Do the authors want to target biologists with segmented data who can use your package and compliment an existing work flow or repeat measurements from another publication? Is it more for a technical audience. Somebody who might use the software as a library or a peice of an existing workflow.

## Conclussion

I cannot really decide who the target audience of this paper would be. Do they have a really strong algorithm they want to publish? It isn't highlighted. Do they have a software tool that somebody should be using to reproduce some specific measurements? That is where a table could help.

Reviewer #2: Review for the manuscript "CellMet: Extracting 3D shape metrics from cells and

tissues", from Sophie Keis and co-authors.

The authors present "CellMet", a software written in Python for the analysis of the

shape of cells and tissues based on 3D images. The introduction presents the context

and the need for quantification of 3D shape. The different sections present the

methods used for obtaining the 3D images used within the manuscript, a description

of the software design and of the features that can be obtained. An applications sections

describes a variety of use cases for the software.

The main originality of the proposed software seemd to be the quantification of the

relationships between cells within 3D images. While this question is of interest for

a variety of applications, the manuscript does not appear to be acceptable for publication

in its present form, for several reasons:

* the title mentions cell shape, and the introduction presents cell morphology. However

most of the proposed metrics concern the arrangement of cells and the topology of the

tissue. This could be more clearly emphasised

* the structure of the manuscript makes it difficult to follow. It is surprising that after

an introduction presenting the software as main subject, the next section are methods presenting

preparation of images. Explaining the meaning of the images in the introduction would clarify.

* The structure of the manuscript does not follow the recommendation of the journal

for "Software" article. Using the recommendation could be an option to clarify the structure.

* The authors use volume and surface area as base metrics, and mention spherical harmonics.

It is however very surprising that they do not mention other "classical" features used for

morphology analysis of 3D regions. Examples are equivalent ellipsoid, thickness, Feret diameters...

While they are not the main topic of the proposed software, they should be mentioned if using

"shape metrics" in the title.

* The authors seem to be totally unaware of the notion of "Region Adjacency Graph", which is

a very classical approach in image processing, and that appears to be the main data structure

of the proposed software.

* finally, in some cases the authors use methods that may provide strong bias, with the aim

of modelling biological processes. This approach can lead to erroneous conclusions, and the

authors must be aware of the possible errors that can be provided by the image analysis methods

they use.

Other comments

* abstract: it could be mentioned in the abstract that the proposed software is written in python

* L144+: it is not very clear how the transition between the 3D label map and the topological data

structure is made. The process of identifying voxel "types" (i.e. within cell, face, edge or vertex)

is clear. However, the process of linking the edges together to consider a curve does not seem to be

explained. Otherwise, I do not understand how the "length" of the edge is computed.

* Also, I strongly suggest dissociating the notion of "half-edge", which a specific implementation,

from the connectivity graph. The authors may also consider the notion of "cell complex", that

is more formally defined, and that can be considered as a generalisation of polygon meshes.

* L155: this is a classical method for building region adjacency graph. At least the term should be

mentioned. Reference papers can also be cited.

* L173: I do not understand the meaning of the last two sentences.

* L178. This paragraph is surprinsingly very short regarding the manuscript title. Authors should

be aware that it is not necessary to resize/resample the image to measure the volume.

* L183: evaluating the surface area by simply counting the number of boundary voxels leads to

a strong overestimation. If the purpose is simple to compare populations of cells, this is fine.

But as the authors aim to link with modelling approaches, this bias should be taken into account

and commented. Check for example Joakim Lindblad & Ingela Nyström

"Surface area estimation of digitized 3D objects using local computations", 2002,

or methods based on discretization of Crofton formula, such as C. Lang, J. Ohser, R. Hilfer,

"On the analysis of spatial binary images", Journal of Microscopy, vol 203, 2001.

* L184: using the maximum value of the distance transform does not yields the centroid, but

the center of the largest inscribed ball or sphere.

* L187: it is not very clear what the author want to quantifiy within this sentence.

* L189: computing an equivalent ellipsoid would provide a 3D orientation, not a series of orientations

along various planes, which is less generic.

* L190: (more a question). Do the author refer to a specific definition for the sphericity?

Common software such as ImageJ use definition for (2D) circularity as 4*pi*A/P^2. I would have expected

a straightforward extension to 3D, but I am not sure a consensual definition exist.

* Fig. 2: the notion of "edge curvature" is IMHO erroneous. The curvature is defined locally,

for each point of the curve (or of a surface). I would rather speak of "tortuosity", or maybe

another term ("straightness"?)

* Fig.4: the authors propose a very specific shape analysis to distinguish between scutoid and prism shapes.

I would have appreciated a comparison with more classical shape features.

* It would be fine to compare the software with other software that allow management and analysis of topological

arrangement of cells. See e.g. "Image Processing Filters for Grids of Cells Analogous to Filters Processing

Grids of Pixels", by Robert Haase, in Frontiers in Computer Sciences (2021),

or "Griottes: a generalist tool for network generation from segmented tissue images", by Gustave Ronteix

in BMC Biology (2022).

**Have the authors made all data and (if applicable) computational code underlying the findings in their manuscript fully available?**

Reviewer #1: Yes

Reviewer #2: Yes

PLOS authors have the option to publish the peer review history of their article (what does this mean?). If published, this will include your full peer review and any attached files.

Reviewer #1: **Yes: **MB Smith

Reviewer #2: No

**Figure resubmission:**
---

## [Decision Letter · Decision Letter 1]

11 May 2025

PCOMPBIOL-D-24-01768R1

CellMet: Extracting 3D shape and topology metrics from confluent cells within tissues

PLOS Computational Biology

Dear Dr. Theis,

Thank you for submitting your manuscript to PLOS Computational Biology. After careful consideration, we feel that it has merit but requires substantial rewriting to fit PLOS Computational Biology's publication criteria as it currently stands. Therefore, we invite you to submit a revised version of the manuscript that addresses the points raised during the review process.

Please submit your revised manuscript within 60 days Jul 11 2025 11:59PM. If you will need more time than this to complete your revisions, please reply to this message or contact the journal office at ploscompbiol@plos.org. Please include the following items when submitting your revised manuscript:

We look forward to receiving your revised manuscript.

Kind regards,

Virginie Uhlmann

Academic Editor

PLOS Computational Biology

Marc Birtwistle

Section Editor

PLOS Computational Biology

**Journal Requirements:**

1) Thank you for stating "All work on Zebrafish was approved by the University of Warwick animal welfare and ethical review board (AWERB, code 77 20-21) and adhered to the Animals (Scientific Procedures) Act 1986, and Home Office ASPeL regulations for animal work." Please insert the Ethics Statement at the beginning of your Methods section in the manuscript.

2) Please ensure that the funders and grant numbers match between the Financial Disclosure field and the Funding Information tab in your submission form. Note that the funders must be provided in the same order in both places as well. Currently, "Warwick startup support" is missing from the Funding Information tab.

3) Please upload the figures in a correct numerical order in the online submission form.

**Reviewers' comments:**

Reviewer's Responses to Questions

Reviewer #1: I feel the authors have adequately addressed my concerns. They do a better job of explaining the purpose of their software. I still feel there are still some issues concerning readablilty. Here are two examples:

Lines 111 to 119. The descriptions of the intermediate outputs (npz, obj, ply) seems to be largely irrelevant at this point. In contrast the statement "As CellMet does not correct for mis-segmentation; this need to be done prior to analysis." to be very important. I think this should be expounded upon, what are mis-segmentations? Each cell needs a unique label, the the labels need to be tightly packed, the epithelia needs to have a apical basil axis along the z-axis. Naively I take an image, I run it through cellpose and get a set of labels. Then I run it through CellMet is that going to work?

Some of the 2D/3D confusion has statements like, lines 86-90. Is it useful for 2D analysis? It doesn't sound like it, but the authors don't want to let it go. Do the authors have code examples and jupyter notebooks that demonstrate 2D analysis? If they do some 2D examples there, then it might be worth including it in a section of it's own. "CellMet performs limited 2D analysis, xyz, found in our examples. Otherwise lines 86-90 should be reduced to "CellMet is not suitable for 2D analysis." Why say "it can do it"? It only weakens the other claims.

The following details, I am a bit confused about, but the other reviewer seemed to be more expert on the topic.

Line 166, I don't understand the point of rotating. I can only guess that it is used in another calculation somewhere, but I don't understand why.

Line 192 Was a PCA analysis performed? They also mention fitting the plane to the points and projecting the points onto the plane, is this part of the PCA analysis? Would it be possible to write the equation?

Reviewer #2: First of all, I would like to thanks the authors for the integration of the different remarks, that

make the manuscript much easier to understand. However, there are still many parts that are

difficult to understand, or that are too technical for the target audience. My main comment is

that authors focus too much on the technical details, making it difficult for the reader to

decide if the software may or not correspond to its needs.

Comments:

* L29-35. This paragraph is very important for understanding the manuscript, and should be emphasised.

In particular, the lack of (user-friendly) software solution for analysing spatial arrangement of cells

can be better discussed. We can find some tools for building region adjacency graphs, for examples, but

this requires to build a complete workflow using scripting and/or programming. Also we can find some

work related to the description of cell arrangement within tissues (see e.g. "Automatic identification

and characterization of radial files in light microscopy images of wood", https://doi.org/10.1093/aob/mcu119,

or "DRACO-STEM: An Automatic Tool to Generate High-Quality 3D Meshes of Shoot Apical Meristem Tissue at

Cell Resolution, https://doi.org/10.3389/fpls.2017.00353).

The notions of edges and faces should be introduced later, in the methods, as they describe a specific

point of view for analysing the data.

* L67: this paragraph is still difficult to understand, and needs rewriting for clarification.

I suggest starting by explaining what features the authors want to quantify, and explaining how they are

quantified in a second time.

Here, the morphology of cells is described through that of the faces and edges they share with their

neighbors. This requires to 1) identify neighbors, 2) use an appropriate data structure to represent

how cells, faces and edges are related.

The notion of half-edge is very technical, and should be introduced later. If it is used to describe

edge/face morphology of individual cells, using "polygon mesh" terminology makes the things much easier

to understand.

* L70: note that the topology is independent of the geometry, so writing that topology data contain shape

information is a kind of non-sense...

* L82: could be useful to explain / define what are prism or scutoid shapes earlier in the manuscript.

As the scutoid seem to be a cell type specific to epithelia tissue, it could be beneficial to have a

short explanation on such tissues (and the typical properties of epithelium cells) within the introduction.

* a side remark on the violing plots of response documents: the plots depict distribution of integer values.

Using violin plot is not the most appropriate for the nature of the data, due to the discrete nature of

integers. A box plot would be better. In particular, as the smoothing factor used for box was chosen too small,

this induces waviness in the representation that are particularly intriguing, and distract the reader from the

main message.

* L225-226: this sentence is an easy to understand summary of the different features the software provides.

I strongly suggest to use it earlier, either in the introduction (typically around L39-40), or at the beginning

of the "design and implementation" section.

**Have the authors made all data and (if applicable) computational code underlying the findings in their manuscript fully available?**

Reviewer #1: Yes

Reviewer #2: Yes

PLOS authors have the option to publish the peer review history of their article (what does this mean?). If published, this will include your full peer review and any attached files.

Reviewer #1: No

Reviewer #2: No

**Figure resubmission:**
---

## [Decision Letter · Decision Letter 2]

22 Jun 2025

Dear Dr Theis,

We are pleased to inform you that your manuscript 'CellMet: Extracting 3D shape and topology metrics from confluent cells within tissues' has been provisionally accepted for publication in PLOS Computational Biology.

Best regards,

Virginie Uhlmann

Academic Editor

PLOS Computational Biology

Marc Birtwistle

Section Editor

PLOS Computational Biology

Reviewer's Responses to Questions

**Comments to the Authors:**

Reviewer #2: Thank you for the revised version of the manuscript. I have read the final version, and all the remarks I had have been answered by the authors. I therefore agree to the publication of the manuscript in its present form.

**Have the authors made all data and (if applicable) computational code underlying the findings in their manuscript fully available?**

Reviewer #2: Yes

PLOS authors have the option to publish the peer review history of their article (what does this mean?). If published, this will include your full peer review and any attached files.

Reviewer #2: No

---

## [Editor Report · Acceptance letter]

PCOMPBIOL-D-24-01768R2

CellMet: Extracting 3D shape and topology metrics from confluent cells within tissues

Dear Dr Theis,

I am pleased to inform you that your manuscript has been formally accepted for publication in PLOS Computational Biology. Your manuscript is now with our production department and you will be notified of the publication date in due course.

With kind regards,

Judit Kozma
